# Anticipated health behaviour changes and perceived control in response to disclosure of genetic risk of breast and ovarian cancer: a quantitative survey study among women in the UK

Susanne F Meisel,[1,2] Lindsay Sarah Macduff Fraser,[3] Lucy Side,[4] Sue Gessler,[3,5] Katie E J Hann,[3] Jane Wardle,[2] Anne Lanceley,[3,5] PROMISE study team

JW was deceased on 20 Oct 2015.

For numbered affiliations see end of article.

**Correspondence to**
Dr Anne Lanceley;
a.lanceley@ucl.ac.uk

## ABSTRACT

**Background** Genetic risk assessment for breast cancer and ovarian cancer (BCOC) is expected to make major inroads into mainstream clinical practice. It is important to evaluate the potential impact on women ahead of its implementation in order to maximise health benefits, as predictive genetic testing without adequate support could lead to adverse psychological and behavioural responses to risk disclosure.

**Objective** To examine anticipated health behaviour changes and perceived control to disclosure of genetic risk for BCOC and establish demographic and person-specific correlates of adverse anticipated responses in a population-based sample of women.

**Design** Cross-sectional quantitative survey study carried out by the UK Office for National Statistics in January and March 2014.

**Setting** Face-to-face computer-assisted interviews conducted by trained researchers in participants' homes.

**Participants** 837 women randomly chosen from households across the UK identified from the Royal Mail's Postcode Address File.

**Outcome measures** Anticipated health behaviour change and perceived control to disclosure of BCOC risk.

**Results** In response to a genetic test result, most women (72%) indicated 'I would try harder to have a healthy lifestyle', and over half (55%) felt 'it would give me more control over my life'. These associations were independent of demographic factors or perceived risk of BCOC in Bonferroni-corrected multivariate analyses. However, a minority of women (14%) felt 'it isn't worth making lifestyle changes' and that 'I would feel less free to make choices in my life' (16%) in response to BCOC risk disclosure. The former belief was more likely to be held by women who were educated below university degree level (P<0.001) after adjusting for other demographic and person-specific correlates.

**Conclusion** These findings indicate that women in the UK largely anticipate that they would engage in positive health behaviour changes in response to BCOC risk disclosure.

## Strengths and limitations of this study

► Stratified random sampling, the 'gold-standard' for survey recruitment, was used in this study.
► The study reports on anticipated responses to genetic breast cancer and ovarian cancer (BCOC) risk information from a large general population sample of women living in the UK.
► The survey questions were broad and may have been interpreted differently by different participants.
► A general hypothetical scenario about receiving genetic BCOC risk information was used instead of asking participants to consider a hypothetical scenario of receiving a high-risk or low-risk genetic test result.

associated with an increased risk of developing breast cancer and ovarian cancer (BCOC). In addition to germline mutations in *BRCA1* and *BRCA2*, which confer a very high lifetime risk of BCOC (39%–65% and 11%–37%, respectively),[1–3] a growing list of moderate-risk and lower-risk markers specific to each cancer type have been identified.[4–10] These influence cancer development in a larger proportion of the population despite small-effect sizes. It is anticipated that, in future, some moderate-risk markers will be included in existing panel tests used in clinical genetics practice.[11–13] While high penetrance genes such as *BRCA1* and *BRCA2* confer a lifetime risk for cancer of 50% or greater, the lifetime risk for moderately penetrant genes ranges from 20% to 50%, and low penetrance genes are thought to have a limited effect.[14]

To date, genetic testing of *BRCA1* and *BRCA2* is conducted in clinical practice only, in individuals with a strong family history or after a cancer diagnosis. However, it has been argued that this approach is likely to miss a

## INTRODUCTION

Advances in genetic technologies have led to the identification of gene mutations

substantial proportion of women at high risk (~30%), particularly if the family is small, the mutation was transmitted through the paternal line or the individual has a number of lower-risk variants whose effects are additive.[15–18] Given that genetic testing for BCOC risk can now be carried out at a relatively low cost, there have been calls to make it available on a population level, based on the premise that timely awareness of genetic risk would empower individuals to make informed decisions about cancer risk management, allow for early intervention and ultimately improve clinical outcomes.[19] There is a drive from healthcare professionals to 'mainstream' genetic testing as part of routine cancer care in oncology services, aimed at improving diagnosis and providing individualised treatment for patients (http://www.mcgprogramme.com/).[20–23] *BRCA1* and *BRCA2* testing may be offered to women diagnosed with triple negative breast cancer, and information on genetic status is likely to influence clinical care and management of the patient and the family.[24 25] Moreover, genetic testing for BCOC susceptibility can already be purchased over the internet[26] (www.23andme.com; www.getcolor.com). The notion that individuals have a 'right to know' their genetic information has also gained traction in recent years.[27–30]

However, concerns persist about patient safety related to poor psychological adjustment following genetic risk disclosure. Studies investigating reactions to disclosure of BRCA status commonly do not find evidence for persistent negative psychological outcomes beyond 1 year after result disclosure.[31–34] There is also some evidence suggesting that genetic test results indicating a higher risk of disease can lead to behaviour change such as uptake of breast and colorectal cancer screening,[31 35 36] although two systematic reviews found no significant impact of genetic risk information on other behaviours[37 38] including smoking, physical activity, use of medication or vitamins, diet, alcohol use and use of a sunscreen. However, genetic testing has hitherto only been performed in women with a strong family history of BCOC, so these results may not generalise to a population largely unaware of any increased risk.

Currently, one-to-one genetic counselling is strongly embedded within clinical genetic testing services, offering education and support throughout the process. If genetic testing were introduced on a population level, the current model of delivering genetic counselling would no longer be feasible. It has been argued that returning genetic risk test results for highly debilitating conditions such as BCOC without appropriate professional guidance would leave room for misinterpretation and uncertainty, increasing the risk of adverse psychological outcomes.[39–42] In recent years, research has begun to investigate new approaches of providing cancer genetic services. Mainstreaming genetic testing, by offering women with a diagnosis of ovarian cancer testing directly through the oncology clinic and providing post-test genetic counselling via clinical genetics for those with a pathogenic variant, is one example.[43 44] Research has found that mainstreaming genetic testing in this way can successfully reduce time and resources spent on the processes[45] and does not appear to negatively affect patients' psychological well-being.[46] Genetic counselling with patients and healthy family members via telephone or video consultations instead of face-to-face genetic counselling has also been found to be feasible, and patients are generally satisfied.[47–49] These new approaches to offering genetic testing and providing genetic counselling for cancer susceptibility may be advantageous in future if testing is offered to patients more widely.

Although a wealth of studies have investigated anticipated psychological reactions to disclosure of BCOC genetic risk in higher-risk groups,[31 50 51] little is known about this issue in women unselected for family history. One qualitative study found that women in the UK would support the idea of genetic testing for risk of ovarian cancer because they expected it to trigger positive lifestyle changes and provide a means of control over cancer development.[52] However, in a similar study investigating attitudes towards genetic testing for breast cancer risk in the Netherlands, women were more ambivalent about disclosure of genetic risk; although there was recognition of the benefits of testing, some women were uncertain about their reactions to risk disclosure and the impact it would have on their lives.[53]

An early population-based survey investigated attitudes towards genetic testing for breast cancer among 836 women living in Washington State.[54] Results showed that over three quarters of the sample expected that awareness of their genetic risk of breast cancer would give them more control over their life, and a similar number reported that they would 'definitely' or 'probably' take part in testing if it was offered to them. A more recent population-based study with over 800 participants in the Netherlands compared attitudes towards genetic testing between 2002 and 2010.[55] Results showed that positive expectations about the utility of genetic testing had increased over the 8-year interval. However, about a third of participants believed that awareness of genetic risk would deprive people of the freedom to live as they want, and this proportion remained unchanged over time. Although these findings give an indication about the acceptability of population-based genetic testing, they may be culture-specific and may therefore not generalise to other populations. Furthermore, the latter survey asked only about attitudes towards genetic testing more broadly; attitudes to genetic risk disclosure specific to BCOC may be different.

The aim of the present study was to examine the anticipated health behaviour changes and perceived control in response to disclosure of genetic risk for BCOC and to explore demographic and person-specific correlates (eg, marital status and family history of cancer) of anticipated response among a population sample of women in the UK. Identifying subgroups who are more likely to anticipate a negative response to learning about their genetic risk of developing these cancers, before genetic testing is

integrated into mainstream healthcare (eg, by including it in current cancer screening programmes), gives scope for timely intervention to avoid further increasing disparities in healthcare.

## METHODS AND PROCEDURE

This quantitative study complied with the principles set out in the Declaration of Helsinki. As a population-based anonymous survey, it was exempt from ethical approval, in accordance with the guidelines set out by the University College London ethics committee for non-National Health Service research (http://ethics.grad.ucl.ac.uk/exemptions.php).

### Sample

Data were collected by the UK Office for National Statistics (ONS) in their monthly survey which is conducted on behalf of a range of organisations (eg, government departments, non-governmental institutions and academic departments). Each month, 2010 households are identified from the Royal Mail's Postcode Address File using stratified random probability sampling. Selected addresses are contacted up to eight times at different times and days of the week to maximise response rates. One person aged over 16 years from each household is randomly chosen to complete a computer-assisted face-to-face interview with trained researchers. Surveys commonly include a variety of different topics, making participation bias unlikely, although participants can opt out at any point. Questions on the 'genetics and screening' module were included in data collection waves in January and March 2014.

### Measures

The 'genes and cancer' module was introduced with a short statement: 'Genes contain the 'instruction manual' of life, called DNA. Genes are passed from parents to their children. Nowadays, it is possible to predict whether someone is likely to develop certain diseases by looking at their genes. This is called genetic testing'. No further information was given, as we were interested in women's attitudes towards BCOC genetic testing based on their current understanding of it. Participants were asked several questions in relation to genetic testing for BCOC risk and attitudes on breast cancer screening (see online supplementary material for detail). Some of the findings have previously been published.[56]

### Outcome variables

Anticipated reactions to learning about increased BCOC risk were assessed with statements ranging from positively framed outcomes such as the opportunity to exert personal control ('If I knew my genetic risk of breast cancer and ovarian cancer, it would give me more control over my life') or to pursue health behaviours aimed at mitigating risk ('If I knew my genetic risk of breast cancer and ovarian cancer I would try harder to have a healthy lifestyle'), to more negatively framed outcomes such as

curtailing freedom of choice ('If I knew my genetic risk of breast cancer and ovarian cancer, I would feel less free to make choices in my life') or the perceived futility of attempting behaviour change ('If I knew my genetic risk of breast cancer and ovarian cancer, I would feel it wasn't worth making lifestyle changes'). All items were scored on a 5-point Likert scale ranging from 'strongly disagree' to 'strongly agree' and were adapted from previous research.[54 55]

### Predictor variables

Perceived risk of BCOC was each assessed with one question 'Compared with other women of your age, what do you think are your chances of getting breast [ovarian] cancer' with response options of: 'much lower than others', 'lower than others', 'the same as others', 'higher than others' and 'much higher than others'. Given that individual risk categories pertaining to much lower/much higher perceived risk were small, perceived risk was grouped into three categories: 'much lower/lower than others, 'same as others' and 'higher/much higher than others'.

Age, ethnicity, educational attainment and marital status were derived using standard survey items (https://tinyurl.com/ya6gn2g6). Age was coded as ≤50 years vs >50 years to investigate any generational differences in anticipated reactions and attitudes. We chose 50 years as the cut-off point since public interest in the UK for genetic testing has been reported as being greatest for people aged 46–60 years,[57] and we felt that the age of 50 years, being close to the average age of menopause in the UK (which is 51 years), was a reasonable midpoint to assess generational effects. Ethnicity was classified as 'White' versus 'ethnic minority' because the individual ethnic minority subgroups were small. Educational attainment was classified as university degree or equivalent versus below university degree level. Marital status was coded as married/cohabiting versus single/widowed/divorced. Family history of any cancer was assessed with one question: 'Have your mother, father, or any of your brothers or sisters, been diagnosed with cancer?' with response options being 'yes', 'no' and 'don't know'.

### Statistical analyses

We included women aged 18–74 years to reflect the population to whom genetic testing for risk of BCOC would likely become available in the future. Statistical analyses were carried out using SPSS V.20.0 (SPSS). Descriptive statistics were explored using frequency tables. We explored demographic (age, ethnicity and education) and person-specific predictors (marital status, family history of cancer, perceived risk of ovarian cancer and perceived risk of breast cancer) for each outcome variable using $\chi^2$ tests. Multivariate logistic regression models were used to investigate the demographic and person-specific correlates of anticipated reactions to disclosure of personal BCOC risk. Bonferroni corrections

were employed in all multivariate analyses to correct for multiple testing, with α=0.0125.

## RESULTS
### Demographic and personal characteristics

Data for the current study were collected in two waves: January and March 2014. In the January wave, 8% (n=166) of the 2010 selected households were not eligible because they were businesses or empty properties. Of 1844 eligible households, 9% (n=171) could not be contacted, and 33% (n=608) declined to take part in the ONS survey. In the March wave, 1853 households were eligible (92%). Of those, 13% (n=237) could not be contacted, and 31% (n=578) chose not to take part. Therefore, the overall response rate was 57%, comparable to previous ONS surveys (ONS 2013; ONS 2012; http://www.ons.gov.uk/ons/index.html). The total female sample was n=1095.

After excluding participants with incomplete data for any of the outcome variables (n=120) or who were aged younger than 18 years or older than 74 years (n=138), the final sample for analysis consisted of 837 women. Table 1 presents the demographic and personal characteristics of the sample. In common with other survey studies, education was higher in the sample than in the general population; other demographic characteristics were comparable to the UK population of women aged 18–74 years (ONS Census 2011: http://www.ons.gov.uk/ons/guide-method/census/2011/index).

Although 36.4% (n=305) of the sample reported a family history of any cancer, most women felt that they were not particularly at risk of breast or ovarian cancer; only 9.9% (n=83) thought that they were at 'higher' or 'much higher' than average risk of breast cancer, and even fewer (7.6%, n=64) thought that they were at 'higher' or 'much higher' risk of ovarian cancer. A full breakdown of responses can be found in the online supplementary material table 1.

### Support of genetic testing

Overall, women were supportive of the idea of genetic testing for breast and ovarian cancer. Only 2.0% (n=18) thought that it was 'never' a good time to have a genetic test. However, about a third of the sample (31.4%, n=290) were 'unsure'. Opinions were divided on when the best time would be to get tested, with 8.3% (n=77) thinking that it would be 'just after birth', just over a quarter (27.4%, n=253) thought 'during school', 10.6% (n=98) reported 'just before marriage', 16.9% (n=156) said 'before they plan to have a child' and 3.4% (n=31) thought the best time would be 'when they are unwell'. There were no demographic or personal correlates with 'ever' versus 'never' supporting genetic testing for breast and ovarian cancer (data not shown).

**Table 1** Sample demographic and personal characteristics (n=837)

| | n (%) |
|---|---|
| **Gender** | |
| Female | 837 (100.0) |
| **Age (mean, SD)** | 45.9 (15.5) |
| **Age group** | |
| 18–29 years | 143 (17.1) |
| 30–39 years | 179 (21.4) |
| 40–49 years | 160 (19.1) |
| 50–64 years | 234 (28.0) |
| 65–74 years | 121 (14.5) |
| **Ethnicity** | |
| White British | 725 (86.6) |
| White other | 52 (6.3) |
| Mixed ethnicity | 12 (1.4) |
| Black | 11 (1.3) |
| South Asian | 28 (3.3) |
| Other | 9 (1.1) |
| **Education** | |
| No formal qualifications | 126 (15.1) |
| Less than degree level qualification | 478 (57.1) |
| Degree level (or equivalent) | 233 (27.8) |
| **Marital status** | |
| Married/civil partnership/cohabiting | 459 (54.8) |
| Single | 177 (21.1) |
| Divorced/separated | 140 (16.7) |
| Widowed | 61 (7.3) |
| **Family history of cancer (mother, father, brother and sister)** | |
| Yes | 305 (36.4) |
| No/don't know | 532 (63.6) |
| **Perceived chances of getting ovarian cancer compared with other women of same age** | |
| Much higher/higher | 64 (7.6) |
| About the same as others | 590 (70.5) |
| Much lower/lower | 183 (21.9) |
| **Perceived chances of getting breast cancer compared with other women of same age** | |
| Much higher/higher | 83 (9.9) |
| About the same as others | 616 (73.7) |
| Much lower/lower | 137 (16.4) |

### Anticipated response to disclosure of genetic risk of breast and ovarian cancer

Overall, women anticipated to respond proactively to awareness of their genetic risk. Most (71.9%, n=669) said that they 'would try harder to have a healthy lifestyle', and over half (54.7%, n=506) anticipated that it

would give them 'more control over my life'. However, a minority reported that it would 'not be worth making lifestyle changes' (14.1%, n=130) in response to genetic risk disclosure and that they would feel 'less free to make choices in my life' (16.4%, n=152).

In univariate analyses, there were no differences across demographic and person-specific variables in the expectation that disclosure of genetic risk of BCOC would lead to increased personal efforts 'to have a healthy lifestyle' (table 2). Women who had fewer years of formal education ($\chi^2$ (1)=4.57, P=0.032), were married ($\chi^2$ (1)=4.53, P=0.033) or had a family history of cancer ($\chi^2$ (1)=5.51, P=0.019) were significantly more likely to report that awareness of genetic risk would give them more control over their life (table 2). However, none of these associations remained after adjustment for covariates and multiple testing (table 3).

Women with fewer years of formal education were also significantly more likely to agree that 'it wouldn't be worth making lifestyle changes' in response to genetic risk disclosure ($\chi^2$ (1)=7.17, P=0.007), and this association remained in multivariate analyses (OR 1.99, 95% CI 1.19 to 3.35, P=0.009). Women from ethnic minority backgrounds ($\chi^2$ (1)=4.18, P=0.041), those who perceived themselves at a (much) lower or (much) higher risk of breast cancer ($\chi^2$ (4)=11.08, P=0.026) or ovarian cancer ($\chi^2$ (4)=22.38, P<0.001) were more likely to report that awareness of genetic risk would make them 'feel less free to make choices in their life', but none of these findings were maintained after adjusting for demographic and person-specific factors and for multiple testing (table 3).

## DISCUSSION

This study investigated demographic and person-specific correlates of anticipated health behaviour changes and perceived control in response to disclosure of genetic risk of BCOC in a population-based sample of women in the UK.

Although our results suggested that most women did not expect to be negatively affected by knowing their genetic risk status for BCOC, women who had fewer years of formal education were nearly twice more likely to believe that 'it would not be worth making lifestyle changes' than those who had stayed in formal education for longer. This finding supports some earlier studies into genetic determinism (the idea that genetic make-up is more important than environmental impact for expression of diseases)[58]; although not all studies have found this link.[54 59]

However, the finding that women with fewer years of formal education are less likely to believe that lifestyle changes could be effective for cancer prevention also echoes studies that investigated attitudes towards cancer screening. These studies have found that women with lower socioeconomic status, which is linked to educational attainment and information literacy,[60] were more likely to hold fatalistic attitudes towards cancer development.

Fatalistic attitudes have in turn been implicated in lower levels of engagement with cancer screening.[61 62] It is possible that women who do not believe in the curability of cancer per se are also more likely to report that 'it wouldn't be worth making lifestyle changes' in response to disclosure of BCOC genetic risk, but because we did not assess beliefs about curability of BCOC, this hypothesis remains speculative. Furthermore, research has shown that those with fewer years in education are less aware of cancer risk factors such as poor diet and physical inactivity.[63] A lack of awareness that lifestyle affects a person's risk of cancer would also help to explain why some participants may feel that 'it wouldn't be worth making lifestyle changes'. Although formal years of education do not equate with health literacy, which was not measured in this study, educational attainment is strongly associated with health literacy[64] and can therefore serve as an indicator for health literacy. Given that there are already disparities in engagement with available means of early detection for cancer (eg, screening) across the population, with lower socioeconomic status groups less likely to engage,[65] it will be vital to provide adequate education and information about the nature of genetic testing before it is introduced into mainstream healthcare. This point has also been highlighted by other commentators in the field.[66 67] In addition, it is important that the public has a multifaceted view of cancer and is aware of genetic and non-genetic risk factors and ways to reduce risk despite genetic susceptibility.

Although previous studies in both higher-risk[68 69] and population samples[52 70] have shown that younger women are generally more positive towards genetic testing, we found no age differences in anticipated reactions towards BCOC genetic risk disclosure. It is interesting to speculate about the impact on our study respondents about the benefits of genetic testing as a result of the spike in media coverage that took place shortly before the survey responses were collected at the beginning of 2014. The well-known actress Angelina Jolie publicly announced her reasons for undergoing BRCA genetic testing and talked about how the knowledge that she carried a gene variant had enabled her to choose risk-reducing surgery (double mastectomy) to manage her increased risk of developing breast cancer. This celebrity revelation may have raised awareness of the potential benefits of breast (and ovarian) cancer genetic testing across all age groups,[71 72] which may at least partially account for the finding that older women were as positive about genetic testing as younger women.

We found no association between participants' anticipated response to genetic risk disclosure and perceived risk of either breast or ovarian cancer. However, since few women considered themselves to be at 'higher' or 'much higher' than average risk of BCOC, sample sizes may have been too small to detect associations. Alternatively, these findings may suggest that UK women may have a more realistic understanding of their personal risk of BCOC compared with reports in previous studies from other

**Table 2** Univariate analyses of anticipated reactions to disclosure of genetic risk of breast and ovarian cancer

| Variable | 'I would try harder to have a healthy lifestyle'* | | 'It would give me more control over my life'* | | 'I would feel that it wasn't worth making lifestyle changes'* | | 'I would feel less free to make choices in my life'* | |
|---|---|---|---|---|---|---|---|---|
| | n (%) agree/strongly agree | χ² (df) P value | n (%) agree/strongly agree | χ² (df) P value | n (%) agree/strongly agree | χ² (df) P value | n (%) agree/strongly agree | χ² (df) P value |
| Sample total (n=837) | 611 (73.0) | | 452 (54.0) | | 119 (14.2) | | 143 (17.1) | |
| **Age** | | | | | | | | |
| >50 years | 248 (71.7) | 0.52 (1) | 199 (57.5) | 2.92 (1) | 54 (15.6) | 0.93 (1) | 55 (15.9) | 0.58 (1) |
| ≤50 years | 363 (73.9) | 0.469 | 253 (51.5) | 0.087 | 65 (13.2) | 0.334 | 88 (17.9) | 0.443 |
| **Ethnicity** | | | | | | | | |
| White | 567 (73.0) | 0.004 (1) | 415 (53.4) | 1.52 (1) | 111 (14.3) | 0.04 (1) | 127 (16.3) | 4.18 (1) |
| Ethnic minority | 44 (73.3) | 0.952 | 37 (61.7) | 0.218 | 8 (13.3) | 0.839 | 16 (26.7) | 0.041† |
| **Education** | | | | | | | | |
| University degree | 162 (69.5) | 1.97 (1) | 112 (48.1) | 4.57 (1) | 21 (9.0) | 7.17 (1) | 34 (14.6) | 1.41 (1) |
| Below university degree | 449 (74.3) | 0.160 | 340 (56.3) | 0.032† | 98 (16.2) | 0.007† | 119 (18.0) | 0.234 |
| **Marital status** | | | | | | | | |
| Married/cohabiting | 343 (75.4) | 2.87 (1) | 261 (57.4) | 4.53 (1) | 61 (13.4) | 0.53 (1) | 67 (14.7) | 3.91 (1) |
| Single/widowed/divorced | 268 (70.2) | 0.090 | 191 (50.0) | 0.033† | 58 (15.2) | 0.463 | 76 (19.9) | 0.048† |
| **Family history of any cancer** | | | | | | | | |
| No/don't know | 385 (72.4) | 0.29 (1) | 271 (50.9) | 5.51 (1) | 73 (13.7) | 0.29 (1) | 98 (18.4) | 1.84 (1) |
| Yes | 226 (74.1) | 0.587 | 181 (59.3) | 0.019† | 46 (15.1) | 0.588 | 45 (14.8) | 0.175 |
| **Perceived risk of OC (n=837)** | | | | | | | | |
| Much lower than others | 40 (66.7) | 6.92 (4) | 30 (50.0) | 4.08 (4) | 11 (18.3) | 1.26 (4) | 22 (36.7) | 22.38 (4) |
| Lower than others | 94 (76.4) | 0.140 | 66 (53.7) | 0.394 | 17 (13.8) | 0.867 | 25 (20.3) | <0.001† |
| The same as others | 427 (72.4) | | 314 (53.2) | | 83 (14.1) | | 84 (14.2) | |
| Higher than others | 48 (81.4) | | 39 (66.1) | | 7 (11.9) | | 10 (16.9) | |
| Much higher than others | 2 (40.0) | | 3 (60.0) | | 1 (20.0) | | 2 (40.0) | |
| **Perceived risk of BC (n=837)** | | | | | | | | |
| Much lower than others | 16 (64.0) | 3.58 (4) | 8 (32.0) | 5.17 (4) | 6 (24.0) | 3.46 (4) | 8 (32.0) | 11.08 (4) |
| Lower than others | 88 (78.6) | 0.466 | 61 (54.5) | 0.269 | 14 (12.5) | 0.484 | 27 (24.1) | 0.026† |
| The same as others | 444 (72.0) | | 337 (54.6) | | 84 (13.6) | | 92 (14.9) | |
| Higher than others | 55 (75.3) | | 41 (56.2) | | 13 (17.8) | | 13 (17.8) | |
| Much higher than others | 8 (80.0) | | 5 (50.0) | | 2 (20.0) | | 3 (30.3) | |

*Each statement preceded by: 'If I knew my risk of breast and ovarian cancer, I would…'.
†Significant at P<0.05.
BC, breast cancer; OC, ovarian cancer.

**Table 3** Multivariate logistic regression of demographic and person-specific predictors of anticipated reactions to disclosure of genetic risk of breast and ovarian cancer

| Variable (n=837) | 'I would try harder to have a healthy lifestyle'* | | | It would give me more control over my life'* | | | 'I would feel it wasn't worth making lifestyle changes'* | | | 'I would feel less free to make choices in my life'* | | |
|---|---|---|---|---|---|---|---|---|---|---|---|---|
| | OR | 95% CI | P value | OR | 95% CI | P value | OR | 95% CI | P value | OR | 95% CI | P value |
| **Age** | | | | | | | | | | | | |
| >50 years | 1 | | | 1 | | | 1 | | | 1 | | |
| ≤50 years | 1.86 | 0.84 to 1.66 | 0.321 | 0.80 | 0.60 to 1.11 | 0.205 | 0.99 | 0.64 to 1.53 | 0.988 | 1.22 | 0.81 to 1.85 | 0.333 |
| **Ethnicity** | | | | | | | | | | | | |
| White | 1 | | | 1 | | | 1 | | | 1 | | |
| Ethnic minority | 0.95 | 0.51 to 1.66 | 0.859 | 1.51 | 0.86 to 2.64 | 0.149 | 1.01 | 0.45 to 2.22 | 0.985 | 1.60 | 0.86 to 3.01 | 0.138 |
| **Education** | | | | | | | | | | | | |
| University degree | 1 | | | 1 | | | 1 | | | 1 | | |
| Below university degree | 1.35 | 0.96 to 1.91 | 0.081 | 1.39 | 1.01 to 1.91 | 0.039 | 1.99 | 1.19 to 3.35 | 0.009† | 1.32 | 0.85 to 2.03 | 0.333 |
| **Marital status** | | | | | | | | | | | | |
| Single/widowed/divorced | 1 | | | 1 | | | 1 | | | 1 | | |
| Married/cohabiting | 1.34 | 0.98 to 1.83 | 0.062 | 1.41 | 1.07 to 1.87 | 0.016 | 0.92 | 0.62 to 1.38 | 0.704 | 0.69 | 0.50 to 1.01 | 0.051 |
| **Family history of any cancer** | | | | | | | | | | | | |
| No/don't know | 1 | | | 1 | | | 1 | | | 1 | | |
| Yes | 1.14 | 0.81 to 1.61 | 0.438 | 1.31 | 0.96 to 1.78 | 0.085 | 1.09 | 0.70 to 1.68 | 0.695 | 0.81 | 0.54 to 1.24 | 0.345 |
| **Perceived risk of OC** | | | | | | | | | | | | |
| Much lower/lower than others | 1 | | | 1 | | | 1 | | | 1 | | |
| The same as others | 1.15 | 0.68 to 1.95 | 0.588 | 0.95 | 0.58 to 1.55 | 0.848 | 0.90 | 0.47 to 1.72 | 0.759 | 0.55 | 0.30 to 0.99 | 0.046 |
| Higher/much higher than others | 1.41 | 0.64 to 3.15 | 0.392 | 1.86 | 0.90 to 3.84 | 0.092 | 0.54 | 0.19 to 1.51 | 0.241 | 0.61 | 0.25 to 1.48 | 0.276 |
| **Perceived risk of BC** | | | | | | | | | | | | |
| Much lower/lower than others | 1 | | | 1 | | | 1 | | | 1 | | |
| The same as others | 0.65 | 0.36 to 1.19 | 0.171 | 1.29 | 0.75 to 2.22 | 0.349 | 0.95 | 0.46 to 1.99 | 0.899 | 0.81 | 0.43 to 1.55 | 0.538 |
| Higher/much higher than others | 0.70 | 0.32 to 1.54 | 0.376 | 0.97 | 0.49 to 1.94 | 0.934 | 1.44 | 0.57 to 3.63 | 0.431 | 0.99 | 0.43 to 2.33 | 0.996 |

*Each statement was preceded by: 'If I knew my risk of breast and ovarian cancer….'.
†Significant after Bonferroni correction at P<0.012.
BC, breast cancer; OC, ovarian cancer.

countries.[73][74] Given that all survey items were closed questions, we could not investigate this finding in more depth, but this could be done in future research.

The strengths of this study are the approach to sampling (the sample was selected using stratified random sampling which is the gold standard of survey recruitment, and, therefore, self-selection is unlikely to have biased current findings) and the statistical adjustment for multiple testing, which gives confidence in the robustness of the findings. However, the study also had important limitations. We only assessed anticipated responses to disclosure of genetic risk of BCOC in general, not whether these differed in response to a hypothetical 'higher' or 'lower' genetic risk. Although all questions have been used in previous research with different samples, with comparable findings, anticipated lifestyle changes and their perceived effectiveness could vary depending on the levels of risk conferred. Further research could explore whether attitudes to genetic risk disclosure of BCOC differ when considering different levels of risk. Second, because of survey constraints, we focused our investigation on a limited set of broad questions which could have been interpreted differently by participants. Questions therefore remain about exactly how respondents anticipated changing their behaviour. Future research could investigate this topic with more in-depth questions.

Naturally, responses to 'real' disclosure of genetic risk information may be less positive than our results suggest; while most participants in this study anticipated making an effort to 'have a healthy lifestyle', research has shown that there is generally a 'gap' between intentions and actual behaviour.[75] In addition, given that many individuals have difficulty in accurately anticipating emotional reactions to life events,[76][77] these findings can only give a broad indication about potential effects of disclosure of BCOC genetic risk. Since none of the questions were open-ended, we could not explore the origins of responses in more detail, but this could be done in future research. Finally, although ethnic distribution of the sample was in line with the overall population in the UK because ethnic minority subgroups were small, we were unable to conduct more detailed analyses to investigate whether attitudes to genetic testing differed among women of different ethnic minority backgrounds. It will be important to explore this in future research, particularly since a recent review found a significant lack of research investigating UK black and minority ethnic (BAME) group awareness and attitudes towards genetic testing for cancer risk.[78] Reports that have involved UK BAME groups highlight low awareness of the availability of genetic services and fear and stigma in relation to cancer genetics.[79][80]

Overall, findings from this study indicate that women in the UK are generally positive about the prospect of population-based genetic testing for risk of BCOC. However, those with fewer years in education may be less positive. Given that there are already existing disparities in uptake of available means of early detection for breast cancer, careful planning is needed to avoid increasing these

further. There are concerns in professional circles that predictive genetic testing without adequate support could lead to adverse psychological and behavioural responses to risk disclosure. Although the risk of psychological harm may in part be mitigated by self-selection into genetic testing,[81] it will be vital to understand the origins of anticipated negative responses and provide adequate education and information about the nature of genetic testing before its integration into mainstream healthcare. This study provides evidence to suggest that before population genetic testing is made available, education is needed that is inclusive of people with different levels of information literacy. Further studies will be needed to evaluate education approaches and their impact on women's choices about whether to participate.

**Author affiliations**
[1]Dept of Psychology, Institute of Psychology, Psychiatry and Neuroscience, King's College London, London, UK
[2]Department of Behavioural Science and Health, Institute of Epidemiology and Health Care, University College London, London, UK
[3]Dept of Women's Cancer, UCL EGA Institute for Women's Health, University College London, London, UK
[4]Clinical Genetics, University Hospital Southampton NHS Trust, Southampton, UK
[5]Gynaecological Cancer Centre, University College London Hospitals (UCLH), London, UK

**Contributors** SFM participated in the study design, analysed the data and drafted the manuscript. KEJH helped with data interpretation and critically revised the manuscript. LS, LSMF, SG, AL and JW conceived the study, participated in the study design and coordination and critically revised the manuscript. All authors except JW who sadly died before final submission have seen and approved the final manuscript.

**Funding** This research was funded by grants from the Eve Appeal (London, UK; grant no 509050) and Cancer Research UK (London, UK; grant no 508007). JW was supported by the Cancer Research UK as part of the CR UK-UCL Cancer Centre.

**Competing interests** SFM and KEJH had financial support from the Cancer Research UK and the Eve Appeal for the submitted work.

**Provenance and peer review** Not commissioned; externally peer reviewed.

**Data sharing statement** Data are available on request from the corresponding author: a.lanceley@ucl.ac.uk.

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
