## [Reviewer comments · BMJ Open]

ARTICLE DETAILS

TITLE (PROVISIONAL)	Anticipated health behaviour changes and perceived control in response to disclosure of genetic risk of breast and ovarian cancer: a quantitative survey study among women in the UK
AUTHORS	Meisel, Susanne; Fraser, Lindsay; Side, Lucy; Gessler, Sue; Hann, Katie; Wardle, Jane; Lanceley, Anne

VERSION 1 – REVIEW

REVIEWER	Marion McAllister Cardiff University UK
REVIEW RETURNED	27-May-2017

GENERAL COMMENTS	This study is novel and timely, reporting on anticipated responses to disclosure of genetic risk of breast and ovarian cancer in the UK in a population based sample. The study is well-designed, with some novel and important findings. The manuscript would benefit from just a few minor amendments as follows: Abstract: 1. Please clarify whether the study is qualitative or quantitative. Introduction: 1. Page 5, para 2, line 24-25: please insert 'genetics' between 'clinical' and 'practice'.2. Page 5, para 2: Although mainstreaming of genetic testing is mentioned later on the Introduction (but only providing the example of ovarian cancer) it would be helpful here to mention mainstreaming of BRCA1/2 testing of women with breast cancer e.g. triple negative breast cancers in oncology services.3. Page 6, para 2, line 40-42: citation missing after sentence starting 'Mainstreaming genetic testing'. Methods: 1. Please clarify at the start of the methods whether the study is qualitative or quantitative.2. Please insert a statement on compliance with the Declaration of Helsinki / ethics approval.3. Page 8-9: It would be helpful in the section on 'Measures' to cross-refer to the survey form, provided as either a figure or supplementary material.4. Page 10, line 12-13: Please justify why 50 years was used as the cut-point to investigate generational differences.
---

	Results: 1. Page 12, para 2: It would have been helpful to know how respondents anticipated changing their lifestyles in response to genetic risk information. This could have been included in the survey, and perhaps ought to be mentioned in the 'Limitations' section (which seems to be missing from the Discussion section). Discussion: 1. Page 11, para 2, lines 52-53: 'which would by definition' - not true as it would depend on what had been decided / agreed in terms of returning unanticipated findings. See ACMG guidance and subsequent controversy regarding return of unanticipated findings. See Green et al. (2013) Genet Med, 15, pp. 565-574; Burke et al. (2013) Genet Med 15(11), doi:10.1038/gim.2013.113; ACMG (2015), Genet Med 17(1): 68-69. This needs a bit of amending to clarify. 2. Page 15, para 3, lines 52-53: replace 'in' with 'than' after 'BCOC' 3. Although there is a set of bullet points on strengths and limitations of the study at the start of the manuscript, I am used to seeing a section in the Discussion part of all research articles highlighting strengths and limitations. Is it a policy of BMJ Open for this just to be included as bullet points at the start? If not, I'd prefer to see this included in the Discussion section, as the bullet points don't highlight the following limitations: (i) the lack of ethnic diversity in the sample and (ii) data were not collected on what anticipated lifestyle changes respondents would make.
--	--

REVIEWER	Jennifer Taber Kent State University, USA
REVIEW RETURNED	07-Jul-2017

GENERAL COMMENTS	This manuscript reports data from a cross-sectional survey in which a representative sample of UK women were asked about their expected responses to receipt of genetic information about their risk of breast and ovarian cancer. Most women were in favor of receiving this information and three-quarters thought they would try to improve their health in response. I appreciate the attempt to anticipate where genetic testing is headed and to assess attitudes about testing in the general population, and the sampling seems well done. My concerns, described below, center around the four items included to assess anticipated responses. 1. The manuscript is framed as assessing women's "anticipated responses." This is a vague term that could encompass affective (e.g., distress, depression), cognitive, motivational, and behavioral responses. The authors assessed a very specific type of response—whether women expected to change their health behaviors and whether they would have control over their life. The more narrow scope of these questions should be made clearer throughout, including in the title, abstract, and introduction. Further, on the first page of the discussion, the survey is justified in part by saying that there are concerns about psychological adjustment following receipt of genetic test results, but the survey does not assess psychological outcomes (psychological outcomes are again emphasized on page 6, line 34 and again on the first line of page 7). 2. The survey questions are vague and without knowing how participants interpreted them, it is difficult to know how to interpret the results. First, the authors acknowledge (and state as a limitation)
--

that the questions ask women to anticipate responses to learning of risk independent of whether they learn of high or low risk. It is possible that likelihood of changing behavior, or feeling like one has choices, would differ dramatically according to the risk conferred. Second, the questions refer to having a “healthy lifestyle” and “lifestyle changes” and participants may have interpreted these phrases in a variety of ways. Are diet and exercise medically advisable for women at high risk of breast cancer or ovarian cancer? Or are screening and perhaps prophylactic surgery all that would be recommended? This issue is not covered in the introduction or discussion. Without knowing how participants interpreted these phrases, or whether lifestyle changes would be advised (which would likely depend also on degree of risk conferred), interpreting the results is difficult. Participants may have selected a response because “unsure” was not an option but these responses may not meaningfully reflect their beliefs. What (if any) was the label for the midpoint of the scales for the anticipated response items that ranged from strongly disagree to strongly agree? It may be useful to report the proportion that selected the middle option for each question versus disagreement.

3. The term “personal predictors” is vague and not very informative.

4. In the introduction, it would be useful to have a sense of the degree of risk conferred by moderate- and lower-risk markers.

5. Why treat perceived risk as categorical for some analyses rather than conducting a test in which it is treated as continuous?

6. On page 13 in the discussion, the statement that “women in the UK do not seem to hold particularly strong views about” whether whole genome sequencing should be included in routine new-born screening is an extrapolation that is not supported by the data.

7. Table 2 should include statistics in addition to the p values (chi square values and DF).

8. In Table 2, why is the full sample size shown as 925 when it is stated as 837 elsewhere?

9. The abstract should include the year the data were collected.

REVIEWER	Sarah Knerr University of Washington, United States
REVIEW RETURNED	07-Jul-2017

GENERAL COMMENTS	This manuscript describes results from a cross-sectional in person survey of adult women in the UK that asked women about genetic testing to provider breast and ovarian cancer risk information. The study's strength is the relatively large and generalizable sample of female respondents. Additional weaknesses beyond those highlighted in the manuscript are use of single questions with unknown psychometric properties/measurement characteristics as outcome measures and lack conceptual or theoretical framework(s) guiding the study design, analysis, and interpretation. Also addressing the large proportion of women who responded that they were unsure about the timing of genetic testing and what that means for the study's conclusions about overall "support for genetic testing". Major suggested revisions:  -Additional specificity about the relationship between the study aims, outcomes of interest, and the questions used to measures those outcomes (and their limitations) needed throughout. Specifically, I don't think that the paper makes a strong enough case that a question about the timing of a genetic test is a robust indicator of whether women are supportive of the idea of genetic testing. This outcome (support of genetic testing) isn't included in the study aims, but leads off the results and discussion sections. The results and discussion also pay a good deal of attention to specific results about the timing of a genetic test, a topic that is not discussed in the introduction. Describing the question as measuring two things (support for genetic testing and also perceptions about timing of genetic testing) is confusing. -The describing the outcome measures/questions about hypothetical behavioral and psychological responses to genetic information as "positive anticipated responses" and "deterministic attitudes" are confusing given women's potential response options, the presentation of results, and also the implication (without a corresponding theoretical justification) that endorsing these attitudes is a good/bad in some way. I would suggest keeping the description of the outcome as close to the wording of the question as possible or referring to them as hypothetical behavioral and psychological responses. -Variable coding needs to be justified and remain consistent throughout all analyses and tables. Add in justification for dichotomizing outcomes and using logistic regression when the outcome variable is categorical. Were outcomes dichotomized in the univariate analysis as well? -Mention descriptive statistics (Table 1) in methods section and include sample sizes for modeling in Table 3. -Language in the results section: replace "effects" with "associations" and "survived" with "remained. Replace "finding was maintained" with "association remained". Minor suggested revisions:
---

	 -Use "variant" and/or "pathogenic variant" instead of mutation or hereditary gene mutation. -If introducing the concept of health care disparities in relationship to the current study, more background and connecting logic needs to be provided. -How are address stratified during sampling? Does this impact the analysis? -Replace "had stayed in education for longer" with "more years of formal education" -Concluding that women in the UK don't hold strong views about the timing of genetic testing based on the question that was asked seems misleading. -Define genetic determinism -Education does not necessarily equate with information literacy, which wasn't measured in the study. -Why is the sample size in Table 2 different than the rest of the study?
--	---

VERSION 1 – AUTHOR RESPONSE

Reviewer: 1

Reviewer Name: Marion McAllister

Institution and Country: Cardiff University, UK

Please state any competing interests or state 'None declared': N/A

Please leave your comments for the authors below

This study is novel and timely, reporting on anticipated responses to disclosure of genetic risk of breast and ovarian cancer in the UK in a population based sample. The study is well-designed, with some novel and important findings. The manuscript would benefit from just a few minor amendments as follows:

Abstract:

1. Please clarify whether the study is qualitative or quantitative.

Response: We have now clarified that this is a quantitative study in the title and the abstract

Introduction:

1. Page 5, para 2, line 24-25: please insert 'genetics' between 'clinical' and 'practice'.

Response: We have now added the word 'genetics'.

2. Page 5, para 2: Although mainstreaming of genetic testing is mentioned later on the Introduction (but only providing the example of ovarian cancer) it would be helpful here to mention mainstreaming of BRCA1/2 testing of women with breast cancer e.g. triple negative breast cancers in oncology services.

Response: We thank the reviewer for this comment. We have now added the following: 'There is a drive from healthcare professionals to 'mainstream' genetic testing as part of routine cancer care in oncology services, aimed at improving diagnosis and providing individualised treatment

for patients (<http://www.mcgprogramme.com/>). BRCA1 and BRCA2 testing may be offered to women diagnosed with triple negative breast cancer and carrier status influences treatment and follow-up choices for patients and their relatives'

3. Page 6, para 2, line 40-42: citation missing after sentence starting 'Mainstreaming genetic testing'.

Response: We have added the missing citation.

Methods:

1. Please clarify at the start of the methods whether the study is qualitative or quantitative.

Response: We have now clarified that this study is quantitative.

2. Please insert a statement on compliance with the Declaration of Helsinki / ethics approval.

Response: We have now added the following details:

This quantitative study complied with the principles set out in the declaration of Helsinki. As a population-based anonymous survey, it was exempt from ethical approval, in accordance with the guidelines set out by the UCL Ethics committee for non-NHS research (<http://ethics.grad.ucl.ac.uk/exemptions.php>).

3. Page 8-9: It would be helpful in the section on 'Measures' to cross-refer to the survey form, provided as either a figure or supplementary material.

Response: Thank you for this comment. We have now added the relevant questions from the survey form as supplementary material.

4. Page 10, line 12-13: Please justify why 50 years was used as the cut-point to investigate generational differences.

Response: We thank the reviewer for this comment. We have now included the following:

We chose 50 years of age as the cut-off point since public interest in the UK for genetic testing has been reported as being greatest for people aged 46-60 years and we felt that the age of 50, being close to the average age of menopause in the UK (which is 51), was a reasonable mid-point to assess generational effects.

Results:

1. Page 12, para 2: It would have been helpful to know how respondents anticipated changing their lifestyles in response to genetic risk information. This could have been included in the survey, and perhaps ought to be mentioned in the 'Limitations' section (which seems to be missing from the Discussion section).

Response: We thank the reviewer for this comment. We have now added the following in the limitations section: Secondly, because of survey constraints, we focused our investigation on a limited set of potential enablers and barriers of genetic testing, namely perceived benefits and deterministic attitudes, and were unable to further investigate how exactly respondents anticipated changing their behaviour.

Discussion:

1. Page 11, para 2, lines 52-53: 'which would by definition' - not true as it would depend on what had been decided / agreed in terms of returning unanticipated findings. See ACMG guidance and subsequent controversy regarding return of unanticipated findings. See Green et al. (2013) *Genet Med*, 15, pp. 565-574; Burke et al. (2013) *Genet Med* 15(11), doi:10.1038/gim.2013.113; ACMG (2015), *Genet Med* 17(1): 68-69. This needs a bit of amending to clarify.

Response: We thank the reviewer for this comment. We have now rephrased this sentence.

2. Page 15, para 3, lines 52-53: replace 'in' with 'than' after 'BCOC'

Response: We have made the suggested change.

3. Although there is a set of bullet points on strengths and limitations of the study at the start of the manuscript, I am used to seeing a section in the Discussion part of all research articles highlighting strengths and limitations. Is it a policy of BMJ Open for this just to be included as bullet points at the start? If not, I'd prefer to see this included in the Discussion section, as the bullet points don't highlight the following limitations: (i) the lack of ethnic diversity in the sample and (ii) data were not collected on what anticipated lifestyle changes respondents would make.

Response: We have included both strengths and limitations in the discussion section and have included the points the reviewer suggested.

Reviewer: 2

Reviewer Name: Jennifer Taber

Institution and Country: Kent State University, USA

Please state any competing interests or state 'None declared': None declared

Please leave your comments for the authors below

This manuscript reports data from a cross-sectional survey in which a representative sample of UK women were asked about their expected responses to receipt of genetic information about their risk of breast and ovarian cancer. Most women were in favor of receiving this information and three-quarters thought they would try to improve their health in response. I appreciate the attempt to anticipate where genetic testing is headed and to assess attitudes about testing in the general population, and the sampling seems well done. My concerns, described below, center around the four items included to assess anticipated responses.

1. The manuscript is framed as assessing women's "anticipated responses." This is a vague term that could encompass affective (e.g., distress, depression), cognitive, motivational, and behavioral responses. The authors assessed a very specific type of response—whether women expected to change their health behaviors and whether they would have control over their life. The more narrow scope of these questions should be made clearer throughout, including in the title, abstract, and introduction. Further, on the first page of the discussion, the survey is justified in part by saying that there are concerns about psychological adjustment following receipt of genetic test results, but the survey does not assess psychological outcomes (psychological outcomes are again emphasized on page 6, line 34 and again on the first line of page 7).

Response: We thank the reviewer for this comment. We have now changed the title to reflect the scope and focus of our questions throughout the manuscript.

2. The survey questions are vague and without knowing how participants interpreted them, it is difficult to know how to interpret the results. First, the authors acknowledge (and state as a limitation) that the questions ask women to anticipate responses to learning of risk independent of whether they learn of high or low risk. It is possible that likelihood of changing behavior, or feeling like one has choices, would differ dramatically according to the risk conferred. Second, the questions refer to having a “healthy lifestyle” and “lifestyle changes” and participants may have interpreted these phrases in a variety of ways. Are diet and exercise medically advisable for women at high risk of breast cancer or ovarian cancer? Or are screening and perhaps prophylactic surgery all that would be recommended? This issue is not covered in the introduction or discussion. Without knowing how participants interpreted these phrases, or whether lifestyle changes would be advised (which would likely depend also on degree of risk conferred), interpreting the results is difficult. Participants may have selected a response because “unsure” was not an option but these responses may not meaningfully reflect their beliefs. What (if any) was the label for the midpoint of the scales for the anticipated response items that ranged from strongly disagree to strongly agree? It may be useful to report the proportion that selected the middle option for each question versus disagreement.

Response: We thank the reviewer for this comment. We agree that the survey questions were open to interpretation by participants, but we were unable to include more fine-grained questions because of the scope of the survey. We have now highlighted this limitation in greater detail in the discussion section of the manuscript, and also included this as a major limitation in the ‘bullet points’. We have also included a full breakdown of responses for the outcome variables in the supplementary material.

3. The term “personal predictors” is vague and not very informative.

Response: We have now changed this to person-specific in the manuscript.

4. In the introduction, it would be useful to have a sense of the degree of risk conferred by moderate- and lower-risk markers.

Response: We have now added the following sentence to the 1st paragraph in the Introduction: ‘While high penetrance genes like BRCA1 and BRCA2 confer a lifetime risk for cancer of 50% or greater, the lifetime risk for moderately penetrant genes ranges from 20% to 50% and low penetrance genes have a limited effect.’

5. Why treat perceived risk as categorical for some analyses rather than conducting a test in which it is treated as continuous?

Response: We thank the reviewer for this comment. Given that individual categories of much lower/much higher perceived risk were very small, we chose to collapse categories as shown, and use a categorical model of analysis. We have now included this justification in the manuscript.

6. On page 13 in the discussion, the statement that “women in the UK do not seem to hold particularly strong views about” whether whole genome sequencing should be included in routine new-born screening is an extrapolation that is not supported by the data.

Response: We thank the reviewer for this comment. We have now removed this section.

7. Table 2 should include statistics in addition to the p values (chi square values and DF).

Response: We have originally not included these values in the interest of space. We have now included these.

8. In Table 2, why is the full sample size shown as 925 when it is stated as 837 elsewhere?

Response: We sincerely apologise for this oversight. The full sample is $n = 837$, and we have corrected the table accordingly.

9. The abstract should include the year the data were collected.

Response: We have now included that the survey data was collected in 2014.

Reviewer: 3

Reviewer Name: Sarah Knerr

Institution and Country: University of Washington, United States

Please state any competing interests or state 'None declared': None declared

Please leave your comments for the authors below

This manuscript describes results from a cross-sectional in person survey of adult women in the UK that asked women about genetic testing to provider breast and ovarian cancer risk information. The study's strength is the relatively large and generalizable sample of female respondents. Additional weaknesses beyond those highlighted in the manuscript are use of single questions with unknown psychometric properties/measurement characteristics as outcome measures and lack conceptual or theoretical framework(s) guiding the study design, analysis, and interpretation. Also addressing the large proportion of women who responded that they were unsure about the timing of genetic testing and what that means for the study's conclusions about overall "support for genetic testing".

Major suggested revisions:

Comment: Additional specificity about the relationship between the study aims, outcomes of interest, and the questions used to measures those outcomes (and their limitations) needed throughout. Specifically, I don't think that the paper makes a strong enough case that a question about the timing of a genetic test is a robust indicator of whether women are supportive of the idea of genetic testing. This outcome (support of genetic testing) isn't included in the study aims, but leads off the results and discussion sections. The results and discussion also pay a good deal of attention to specific results about the timing of a genetic test, a topic that is not discussed in the introduction. Describing the question as measuring two things (support for genetic testing and also perceptions about timing of genetic testing) is confusing.

Response: Thank you for this critique. We agree and have removed this section entirely from the manuscript.

Comment: The describing the outcome measures/questions about hypothetical behavioral and psychological responses to genetic information as "positive anticipated responses" and "deterministic attitudes" are confusing given women's potential response options, the presentation of results, and also the implication (without a corresponding theoretical justification) that endorsing these attitudes is a good/bad in some way. I would suggest keeping the description of the outcome as close to the wording of the question as possible or referring to them as hypothetical behavioral and psychological responses.

Response: We thank the reviewer for this comment. We have now made every effort to stick closely to the question wording rather than extrapolating from it.

Comment: Variable coding needs to be justified and remain consistent throughout all analyses and tables. Add in justification for dichotomizing outcomes and using logistic regression when the outcome variable is categorical. Were outcomes dichotomized in the univariate analysis as well?

Response: We have now justified variable coding as appropriate in the manuscript.

Comment: Mention descriptive statistics (Table 1) in methods section and include sample sizes for modeling in Table 3.

Response: We now refer to the descriptive statistics in the methods section, and have included the sample sizes (n = 837) in Table 3.

Comment: Language in the results section: replace "effects" with "associations" and "survived" with "remained. Replace "finding was maintained" with "association remained".

Response: We have now made the suggested changes in the manuscript.

Minor suggested revisions:

Comment: Use "variant" and/or "pathogenic variant" instead of mutation or hereditary gene mutation.

Response: In the two places where we have used "mutation" (page 5 & page 15) we think that this use is appropriate and we have therefore not made the suggested revision.

Comment: If introducing the concept of health care disparities in relationship to the current study, more background and connecting logic needs to be provided.

Response: Thank you for this suggestion. After some discussion we believe we risk unbalancing the paper if we emphasise health care disparities upfront. Instead we have adjusted the discussion with references (78-80) to assist connections to the current study.

Comment: How are addresses stratified during sampling? Does this impact the analysis?

Response: The Office for National Statistics (ONS) uses stratified random sampling to select addresses. This constitutes the 'gold-standard' of survey sampling. This method makes conclusions drawn from the data more robust than other sampling methods. We feel that we have addressed survey methodology in sufficient detail in the manuscript.

Comment: Replace "had stayed in education for longer" with "more years of formal education"

Response: Thanks we have made this change.

Comment: Concluding that women in the UK don't hold strong views about the timing of genetic testing based on the question that was asked seems misleading.

Thank you, we have edited this aspect of the study.

Comment: Define genetic determinism

Response: We thank the reviewer for this comment. We have now defined genetic determinism.

Comment: Education does not necessarily equate with information literacy, which wasn't measured in the study.

Response: We thank the reviewer for this comment. We have now clarified that information literacy is strongly linked with educational attainment.

Comment: Why is the sample size in Table 2 different than the rest of the study?

Response: We sincerely apologise for this oversight. The sample size is $n = 837$ throughout, and we have now updated Table 2 accordingly.

VERSION 2 – REVIEW

REVIEWER	Marion McAllister Cardiff University, UK
REVIEW RETURNED	27-Sep-2017
GENERAL COMMENTS	The authors have responded well to peer review, and the manuscript is stronger now. My only suggestion is this: Reviewer 3 suggested that the authors 'Use "variant" and/or "pathogenic variant" instead of mutation or hereditary gene mutation'; I did not pick up on this on my first review, but I strongly support the suggestion, although the authors have chosen not to make this change. I think it is important to make the change, since it reflects the language used in contemporary clinical genetics practice.

VERSION 2 – AUTHOR RESPONSE

We thank Marion McAllister and Sarah Knerr for their feedback on wording. We have altered this to reflect contemporary clinical genetics practice as follows:

p.5 'mutations' to 'variants'

p 6 'hereditary gene mutation' to 'pathogenic variant'

p15 'gene mutation' to 'gene variant'.